

# Genome-wide identification and transcriptional profiling of the basic helix-loop-helix gene family in tung tree (*Vernicia fordii*)

Wenjuan Liu[1,2,*], Yaqi Yi[1,2,*], Jingyi Zhuang[1,2], Chang Ge[3], Yunpeng Cao[1,2], Lin Zhang[1,2] and Meilan Liu[1,2]

[1] Key Laboratory of Cultivation and Protection for Non-Wood Forest Trees, Ministry of Education, Central South University of Forestry and Technology, Changsha, Hunan, China
[2] Key Lab of Non-wood Forest Products of State Forestry Administration, College of Forestry, Central South University of Forestry and Technology, Changsha, Hunan, China
[3] School of Urban Design, Wuhan University, Wuhan, Hubei, China
[*] These authors contributed equally to this work.

Corresponding author
Meilan Liu, liumeilan0808@163.com

## ABSTRACT

The basic helix-loop-helix (*bHLH*) transcription factor gene family is one of the largest gene families and is extensively involved in plant growth, development, biotic and abiotic stress responses. Tung tree (*Vernicia fordii*) is an economically important woody oil plant that produces tung oil rich in eleostearic acid. However, the characteristics of the *bHLH* gene family in the tung tree genome are still unclear. Hence, *VfbHLHs* were first searched at a genome-wide level, and their expression levels in various tissues or under low temperature were investigated systematically. In this study, we identified 104 *VfbHLHs* in the tung tree genome, and these genes were classified into 18 subfamilies according to *bHLH* domains. Ninety-eight *VfbHLHs* were mapped to but not evenly distributed on 11 pseudochromosomes. The domain sequences among *VfbHLHs* were highly conserved, and their conserved residues were also identified. To explore their expression, we performed gene expression profiling using RNA-Seq and RT-qPCR. We identified five, 18 and 28 *VfbHLH* genes in female flowers, male flowers and seeds, respectively. Furthermore, we found that eight genes (*VfbHLH29*, *VfbHLH31*, *VfbHLH47*, *VfbHLH51*, *VfbHLH57*, *VfbHLH59*, *VfbHLH70*, *VfbHLH72*) were significant differential expressed in roots, leaves and petioles under low temperature stress. This study lays the foundation for future studies on *bHLH* gene cloning, transgenes, and biological mechanisms.

## INTRODUCTION

Transcription factors (TFs) are important regulatory factors that are generally involved in plant development and abiotic stress responses by binding to cis-acting elements of genes (*Khan et al., 2018*). Basic helix-loop-helix (*bHLH*) TFs are widely found in eukaryotes and are the second largest family of TFs in plants (*Carretero-Paulet et al., 2010*). These

family members all have a conserved bHLH domain with approximately 50~60 amino acid residues. According to the ability to bind DNA, *bHLH* TFs can be divided into two categories: DNA binding and non-DNA binding. Of them, the DNA binding category includes E-box binding (5′-CANNTG- 3′) and non-E-box binding, and the most common binding mode in E-box binding is G-box binding (5′-CACGTG-3′) (*Toledo-Ortiz & Quail, 2003*). With the development of high-throughput sequencing technology, an increasing number of plant *bHLH* families have been discovered and identified, which has greatly accelerated research on the regulation of bHLH protein in plant development and stress responses. Until now, 147 and 167 *bHLH* genes were identified in *Arabidopsis thaliana* and rice (*Oryza sativa*), respectively (*Li et al., 2006*; *Toledo-Ortiz & Quail, 2003*). In woody plants, such as *Malus domestica* (*Yang et al., 2017*), *Amygdalus persica* (*Zhang et al., 2018*), *Ziziphus jujuba* (*Li et al., 2019*), *Camellia sinensis* (*Cui et al., 2018*), *Gossypium hirsutum* (*Lu et al., 2018*), *Ginkgo biloba* (*Zhou et al., 2020*) and *Populus trichocarpa* (*Zhao et al., 2018*), there were 175, 95, 92, 120, 437, 85 and 202 *bHLH* genes, respectively.

In plants, *bHLH* TFs are involved in many physiological processes due to their wide variety of structures and binding proteins. For example, *AtSPATULA* promotes the growth of carpel edges and internal pollen tissue in *Arabidopsis thaliana* (*Irepan Reyes-Olalde et al., 2017*). *AtAMS* plays a crucial role in the regulation of the development of tapetum cells and microspores in anthers in the late stage of meiosis in *Arabidopsis thaliana* (*Lou et al., 2018*). Additionally, the *OsbHLH142* gene regulates the early degeneration of the tapetum during anther development in rice (*Ranjan et al., 2017*), and *SlbHLH22* controls flowering time by activating the expression of *SlSFT* or *SlLFY* genes in *Solanum tuberosum* (*Waseem et al., 2019*). In *Malus domestica*, *MdbHLH3* directly regulates the expression of cytosolic malate dehydrogenase *MdcyMDH* to coordinate carbohydrate partitioning and malate accumulation and directly modulates auxin signaling to control leaf shape in response to local spatial gradients (*Hu et al., 2020*; *Yu et al., 2021*). In rubber (*Hevea brasiliensis*), *Hb_MYC2-1* and *Hb_MYC2-2* may regulate cell differentiation, and *Hb_bHLH1* and *Hb_bHLH2* promote rubber biosynthesis (*Yamaguchi et al., 2020*). In addition, *AtbHLH38* and *AtbHLH39* can be induced by salicylic acid under an iron deficient condition, while *AtbHLH11* can induce the accumulation of salicylic acid in response to an iron deficient environment (*Maurer, Arcos & Bauer, 2014*; *Tanabe et al., 2019*). *OsbHLH138* can activate *TMS5* expression and regulate male fertility under different temperature stress (*Wen et al., 2019*). These researches suggest that *bHLH* TFs play an important role in flower development and low temperature stress in plants.

Tung tree (*Vernicia fordii*), with 50%–60% tung oil in seed, is widely distributed in subtropical areas (*Tan et al., 2011*). Because of its excellent properties, tung oil has been widely used as a drying ingredient in paints, varnishes, coatings, and finishes since ancient times (*Zhang et al., 2014*). Tung oil has attracted global attention in recent years because of production security, environmental concerns, and negative effects of synthetic chemical coatings on human health (*Chen, Chen & Luo, 2012*; *Meininghaus, Gunnarsen & Knudsen, 2000*). Tung tree is suitable to grow in place with sufficient sunlight and fertile soil, and they grow poorly under cold. Under different temperature stresses, the growth of tung tree seedlings slowed, and the physiological function of their leaves declined (*Zhang et*

*al., 2020*). In addition, abnormal development of female flowers is one of the causes of low yield (*Liu et al., 2019*). Due to its economic interest as tung oil, the tung tree genome and transcriptome were recently sequenced (*Liu et al., 2019*; *Zhang et al., 2019*), laying a strong foundation for systematic comprehensive analysis of the *bHLH* gene family. The purposes of our study were to identify the tung tree *bHLH* gene family members, to compare their phylogenetic relationships with *Arabidopsis thaliana*, to analyze their gene structures, cis-regulatory elements, tissue expression patterns, as well as expression profiles under low temperature stress in young plantlets, and finally to provide new insights into understanding of molecular evolution and function of *bHLH* genes in tung tree. The results provide valuable clues to further reveal the role of this family in the growth and development of tung tree.

## MATERIALS AND METHODS

### Plant materials and treatment

The flowers and seeds of the tung tree used in this study were cultivated in the experimental area of the Central South University of Forestry and Technology (Qingping Town, Yongshun County, Hunan Province). Flowers were collected from an 8-year-old tung tree 'Putaotong', including male and female flowers at stage 2 (X1, C1) at 30 days before flowering (DBF), stage 4 (X2, C2) at 20 DBF, stage 6 (X3, C3) at 10 DBF and stage 7 (X4, C4) at 1 DBF. The seeds were collected from an 8-year-old tung tree 'Putaotong', including seeds at 10 weeks after flowering (WAF), 15 WAF, 20 WAF, 25 WAF and 30 WAF.

The seeds were sterilized with 0.5% potassium permanganate for 30 min before being stored in sand, and the sand was kept moist until the seeds germinated. Young plantlets with two young leaves were transplanted separately into pots of the same volume and size, and cultured in an artificial climate chamber at 28 °C. The young plantlets were moved to an artificial climate chamber at 4 °C until they had grown to four leaves. The roots, leaves and petioles were collected with liquid nitrogen at 0 h, 2 h, 4 h, 8 h, 12 h, 72 h, 96 h, and 144 h after 4 °C treatment.

### Identification and protein structure analysis of *VfbHLHs*

The genome sequencing of tung tree were obtained from the NCBI (BioProject: PRJNA503685). The hidden Markov model (HMM) file of the *bHLH* domain (PF00010) was downloaded from the Pfam database (http://pfam.xfam.org/), and HMMER v3.0 software was used to find the VfbHLH protein sequences in the tung tree genome (*Zhang et al., 2019*). To further confirm our sequences, we used the online pfam-search tool (http://pfam.xfam.org/search#tabview=tab1) and the SMART tool (http://smart.embl-heidelberg.de/) to screen sequences. We excluded truncated and false genes in our analysis. The conserved motifs of VfbHLH proteins were detected by MEME (http://meme-suite.org/). The hidden code model was constructed by using online WebLogo 3 software.

## Phylogenetic tree construction and the chromosomal locations of *VfbHLHs*

The amino acid sequences of *Arabidopsis* were downloaded from The Arabidopsis Information Resource (TAIR) database (http://www.arabidopsis.org/). Multiple sequence alignment of tung tree and *Arabidopsis* was analyzed by using ClustalW in MEGA X. A phylogenetic tree of tung tree and *Arabidopsis* was constructed based on their conserved domains. We used MEGA X software and the neighbor-joining statistical method (1,000 bootstrap replicates) to construct a rooted phylogenetic tree (*Hall, 2013*). We obtained the evolutionary distances with the p-distance method, and these distances were used to estimate the number of amino acid substitutions per site. The reliability of the phylogenetic tree was established by conducting 1,000 bootstrap sampling iterations.

The *VfbHLH* gene sequences were used as query sequences in BLASTN searches against the tung tree genome to determine the chromosomal location of the *VfbHLH* genes. Each *VfbHLH* gene was mapped to the tung tree genome according to its genome coordinates. The duplicated *VfbHLH* gene segments were confirmed by searching the tung tree genome duplication database (*Zhang et al., 2019*). *VfbHLH* gene mapping and duplicated gene pairs were performed using Tbtools (*Chen et al., 2020*).

## Gene structure and promoter cis-acting regulatory element analysis of *VfbHLHs*

We used the website GSDS (http://gsds.cbi.pku.edu.cn/) to predict the number of exons from the coding domain sequences (CDS) and DNA sequences of the *VfbHLH* genes (*Guo et al., 2007*). The region upstream of the 1,500 bp region at each member's start codon was identified as the sequence of promoters. We used Plantcare (http://bioinformatics.psb.ugent.be/webtools/plantcare/html/) to analyse the *VfbHLH* genes cis-acting regulatory elements.

## Gene expression analysis of *VfbHLHs*

Transcriptomics data of flowers were obtained from the NCBI SRA database (Accession: SRX3843588; SRS3089151; SRS3089154; SRX3843589; SRS3089148; SRS3089147; SRS3089150 and SRX3843585). Transcriptomics data of seeds were obtained from the NCBI SRA database (SRX4488507, SRX4488514, SRX4488515, SRX4488516 and SRX4488517). RNA sequencing was performed by Illumina Hiseq 2000 (Illumina, United States). Each transcriptome had three biological replicates. The number of all mapped reads for each *VfbHLH* gene were counted and normalized into the Fragments Per Kilobase of transcript per Million fragments mapped (FPKM) (*Florea, Song & Salzberg, 2013*). For convenience, the gene expression was based on logarithm base 10 per million fragments ($\log_{10}$ FPKM), and R software (Version 4.2.1; *R Core Team, 2022*) used to standardize values. The statistical power of this experimental design, calculated in RNASeqPower is in Table S7.

Total RNA was extracted using the RNAprep Pure Plant Kit SK1322 (Sangon Biotech, Shanghai, China) according to the manufacturer's protocol. The RNA concentration and purity were checked with agarose gel electrophoresis. First-strand cDNA was synthesized with a HiScript II Q RT SuperMix for qPCR (+gDNA wiper) (Vazyme, Nanjing, China). The cDNA was used as the template for gene expression anaylsis.

Gene relative expression was detected by RT-qPCR. The primers used in this study were listed in Table S1. All primers were synthesized by Hunan Qingke Biotechnology Co., Ltd. Using a Bio-Rad CFX96 Real Time PCR system with SYBR Premix ExTaq II (Takara, Japan) to detect relative expression levels with three replicates. Tung tree elongation factor 1-$\alpha$ ($EF1\alpha$) was used as the internal control (*Han et al., 2012*). The relative expression levels were calculated using the $2^{-\Delta\Delta CT}$ method (*Livak & Schmittgen, 2001*). The significance of data was analyzed by ANOVA (Analysis of Variance) of IBM SPSS Statistics 25 software, and the Origin 2019 software was used for mapping.

## RESULTS

### Genome-wide identification of *VfbHLH* genes in tung tree

On the basis of HMMER search results, we identified 104 bHLH proteins encoded in the tung tree genome. They were named from *VfbHLH1* to *VfbHLH104* according to their order in the tung tree genomic sequence (Table S2). The presence of the *bHLH* domain was confirmed for all identified sequences by checking the Entrez Conserved Domains Database. The *bHLH* domain alignment of 104 *VfbHLHs* showed that 21 amino acid residues (His-10, Glu-14, Arg-15, Arg-17, Arg-18, Ile-22, Asn-23, Arg-25, Leu-34, Leu-37, Val-38, Pro-39, Lys-49, Lys-52, Ala-53, Leu-56, Ala-59, Ile-60, Tyr-62, Lys-64, Leu-66) in their *bHLH* domains were conserved with a consensus ratio greater than 50% (Fig. 1A). All of these conserved residues were consistent with previous studies (*Li et al., 2006*; *Toledo-Ortiz & Quail, 2003*). The Leu-66, the base region of 102 bHLH proteins, was conserved in all bHLH proteins, suggesting that this residue played an important role in promoting *bHLH* dimer formation (*Atchley, Terhalle & Dress, 1999*).

### Phylogenetic analysis and chromosomal location of *bHLH*

The phylogenetic tree of the tung tree and *A. thaliana* bHLH proteins was constructed by aligning multiple domain sequences (Fig. 1B). A phylogenetic tree of *bHLH* genes of tung tree and *Arabidopsis* was established, and *VfbHLHs* were divided into 18 subfamilies named from A to R subfamilies (Fig. 1B). Except for the H and I subfamilies, all subfamilies corresponded to *Arabidopsis*. The H subfamily included the 13, 14 and 21 subfamilies of *Arabidopsis*, and the I subfamily included the 4 and 11 subfamilies of *Arabidopsis*.

Among the 104 *VfbHLH* genes, 98 were mapped to 11 pseudochromosomes in the tung tree genome (Fig. 1C, Table S2), and only six genes were located on scaffolds. Interestingly, over 51.9% of *VfbHLH* genes were located on Chr1 (13 genes), Chr6 (13 genes), Chr9 (10 genes) and Chr10 (18 genes). Furthermore, intraspecies synteny analysis showed that there were two duplicated gene pairs (*VfbHLH2-VfbHLH3* and *VfbHLH93-VfbHLH94*) on the same chromosome, which may be derived from a tandem duplication. There were 12 duplicated gene pairs (*VfbHLH4-VfbHLH74*, *VfbHLH5-VfbHLH76*, *VfbHLH7-VfbHLH72*, *VfbHLH20-VfbHLH81*, *etc.*) on the different chromosomes, which indicated that these gene duplications may derive from chromosome segmental duplication or a large-scale duplication event. A previous study analyzed repeated events in rice and *Arabidopsis* (*Li et al., 2006*), indicating that some *VfbHLH* subfamily members were most likely derived from repetitive events.

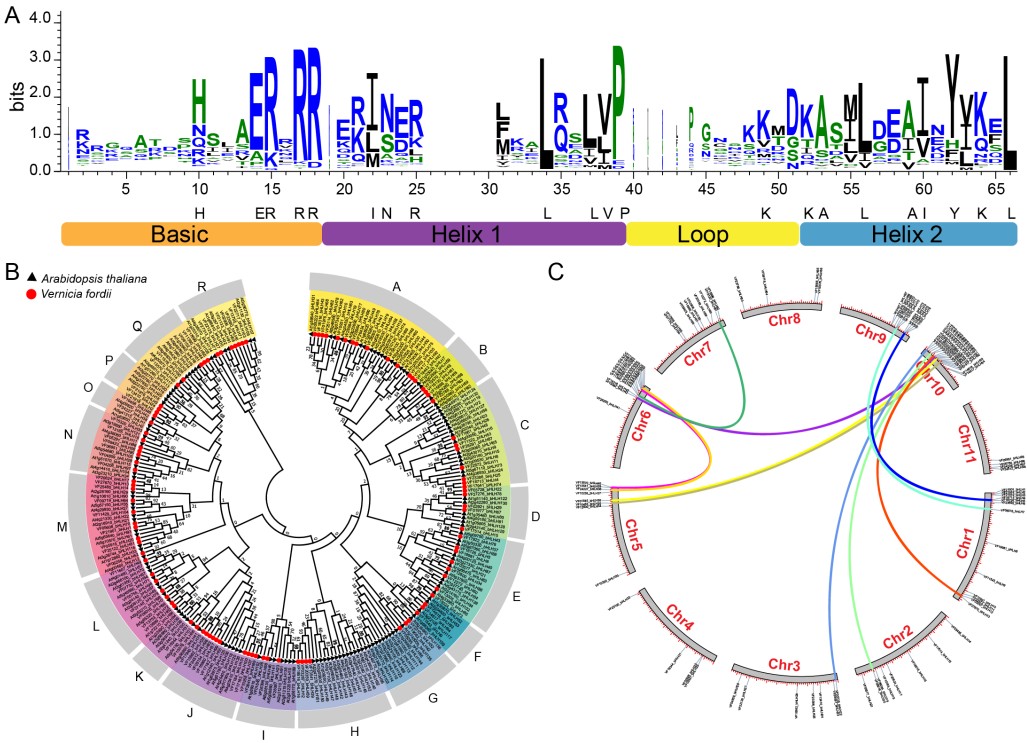

**Figure 1 bHLH domain, phylogenetic relationships, chromosomal distribution and gene duplications of tung tree bHLHs.** (A) The *bHLH* domain is highly conserved across all VfbHLH proteins; The overall height of each stack represents the conservation of the sequence at that position, and capital letters indicate over 50% conservation of amino acids among the 104 *VfbHLH* domains; (B) The phylogenetic analysis of tung tree and *Arabidopsis*. The phylogenetic tree was constructed according to the neighbor-joining method after sequences were aligned with ClustalW using MEGA X with 1,000 bootstrap copies; A–R stand different subfamilies; (C) Chromosomal distribution and gene duplications of *VfbHLHs*. The duplicated gene pairs are linked by different colour lines.

## Conserved motifs and gene structure of *VfbHLHs*

An analysis of the conserved motifs within VfbHLH proteins with the MEME program resulted in the detection of 10 motifs (Fig. 2A). Motifs 1 and 2 were located in HLH domains. Motifs 9 and 10 were located in the bHLH-MYC-N domain. The HLH domain motifs 1 and 2 were highly conserved among the 91 proteins, and only 12 proteins contained motif 1 or 2. In addition, the bHLH-MYC-N domain motifs 9 and 10 were highly conserved among the 14 proteins contained. We also analyzed the intron/exon structures of *VfbHLH* genes (Fig. 2B, Table S3). Most of them had 1∼13 introns. Only nine genes (*VfbHLH5/8/9/40/50/51/55/60/70*) lacked an intron. These nine genes were mainly located in the E, F and Q subfamilies. The differences in the characteristics of the VfbHLH proteins suggest that they are functionally distinct.

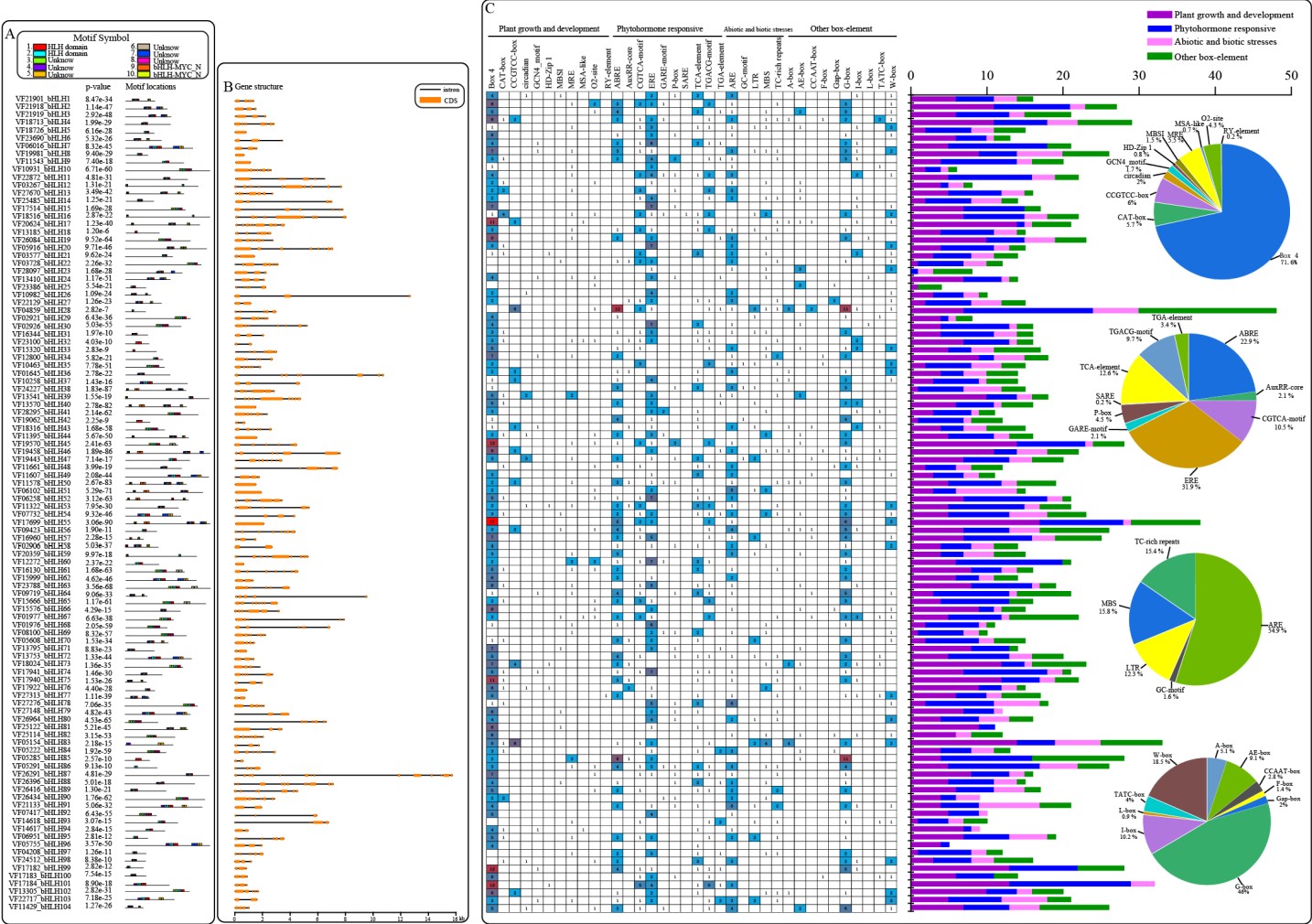

**Figure 2** **Converse motifs, gene structure, and promoter cis-elements of tung tree *bHLHs*.** (A) Conserved motifs of *bHLH* family members are indicated, with each motif represented by a number in a coloured box. Box lengths correspond to motif lengths; (B) The gene structure of *VfbHLH* s. Introns and exons are represented by black lines and orange boxes respectively. (C) Investigation of cis-acting element numbers in all *bHLH* genes. The different colors and numbers of the grid indicate the numbers of different promoter elements in these *bHLH* genes. The different colored histogram represents the sum of the cis-acting elements in each category. Pie charts of different sizes indicate the ratio of each promoter element in each category, respectively.

## Cis-acting element analysis of *VfbHLHs*

Cis-regulatory elements play important roles in regulatory networks controlling plant growth and development, including multi stimulus-responsive genes, and the tissue-specific or stress-responsive expression profiles of genes were closely linked to cis-elements in their promoter regions. Using the PlantCARE database, we identified four categories of cis-elements, including plant growth and development, phytohormone responses, biotic and abiotic stress responses, and other important box-elements in the promoter regions (Fig. 2C). Interestingly, the CCGTCC-box for meristem expression and GCN4-motif required for endosperm expression were found in *VfbHLHs*. Besides, the most common

motif of the phytohormone responsive category was the ERE for cis-acting elements associated with ethylene-responsiveness, accounting for 31.9% of the scanned hormone responsive motifs. Furthermore, various abiotic stress-related elements, such as ARE (anaerobic induction), LTR (low temperature responsive), MBS (drought-inducibility), TC-rich repeats (defence and stress responses) and GC-motif (anoxia), were observed in *VfbHLHs*. Our data suggested that *bHLH* genes of tung tree might play an important role in plant development and abiotic stress responses.

### *VfbHLHs* involved in male and female flower development

To functionally characterize the *VfbHLH* genes, we examined the corresponding expression patterns based on transcriptomic data. We analyzed the following four representative male and female flower developmental stages: stage 2 (X1, C1): 30 DBF; stage 4 (X2, C2): 20 DBF; stage 6 (X3, C3): 10 DBF; stage 7 (X4, C4): 1 DBF. We applied the FPKM value determined via transcriptome profiling to generate a heatmap for the *VfbHLH* expression patterns in developing male and female flowers (Fig. 3A). The *VfbHLH* genes with FPKM values less than one in all samples were considered to be barely expressed. Thus, 16 unexpressed genes were excluded from the heatmap (Table S4). The remaining 98 *VfbHLH* genes used in the heatmap were expressed in at least one sample (Table S4).

To identify genes that were closely associated with a particular developmental stage, we manually screened the expression patterns of *VfbHLH* genes. We defined *VfbHLH* genes with expression levels in one stage that were at least 2-fold greater or more than those in the remaining three stages as "stage-specific" genes (*Feng et al., 2017*). Under such conditions, only one gene (*VfbHLH42*) was considered to be specifically expressed at stage 2 of male flowers, while there were no stage-specific genes in female flowers at stage 2. In stage 4, two genes (*VfbHLH76* and *VfbHLH100*) were considered to be specifically expressed in male flowers and only one gene (*VfbHLH7*) was considered to be specifically expressed in female flowers. In stage 6, there were four (*VfbHLH24/93/94/99*) and one (*VfbHLH60*) stage-specific genes in male and female flowers respectively. In stage 7, male flowers had 11 stage-specific genes, such as, *VfbHLH10/19/30/31/37/43/56/64/68/78/90*, while only three genes (*VfbHLH40/50/57*) were considered to be specifically expressed in female flowers. In addition, RT-qPCR was applied to validate the expression of four stage-specific genes at different developmental stages of male and female flowers (Figs. 3B–3E). These results suggested that stage-specific genes in different stages of male and female flowers might be involved in flower development of tung tree.

### Expression patterns of *VfbHLHs* in tung seed

Tung seed contains 50%–60% tung oil (*Tan et al., 2011*), which can be used to synthesize excellent thermosetting polymers and resins and is considered as a potential source of biodiesel (*Liu et al., 2016*; *Park et al., 2008*). To explore the expression patterns of *VfbHLHs* in tung seed, we analyzed the following five representative seed development stages: 10 WAF, 15 WAF, 20 WAF, 25 WAF, 30 WAF (*Zhang et al., 2019*). During tung seed development, 25 unexpressed genes were excluded from the heatmap and the remaining 79 *VfbHLH* genes used in the heatmap were expressed in at least one stage (Fig. 4A, Table S5).

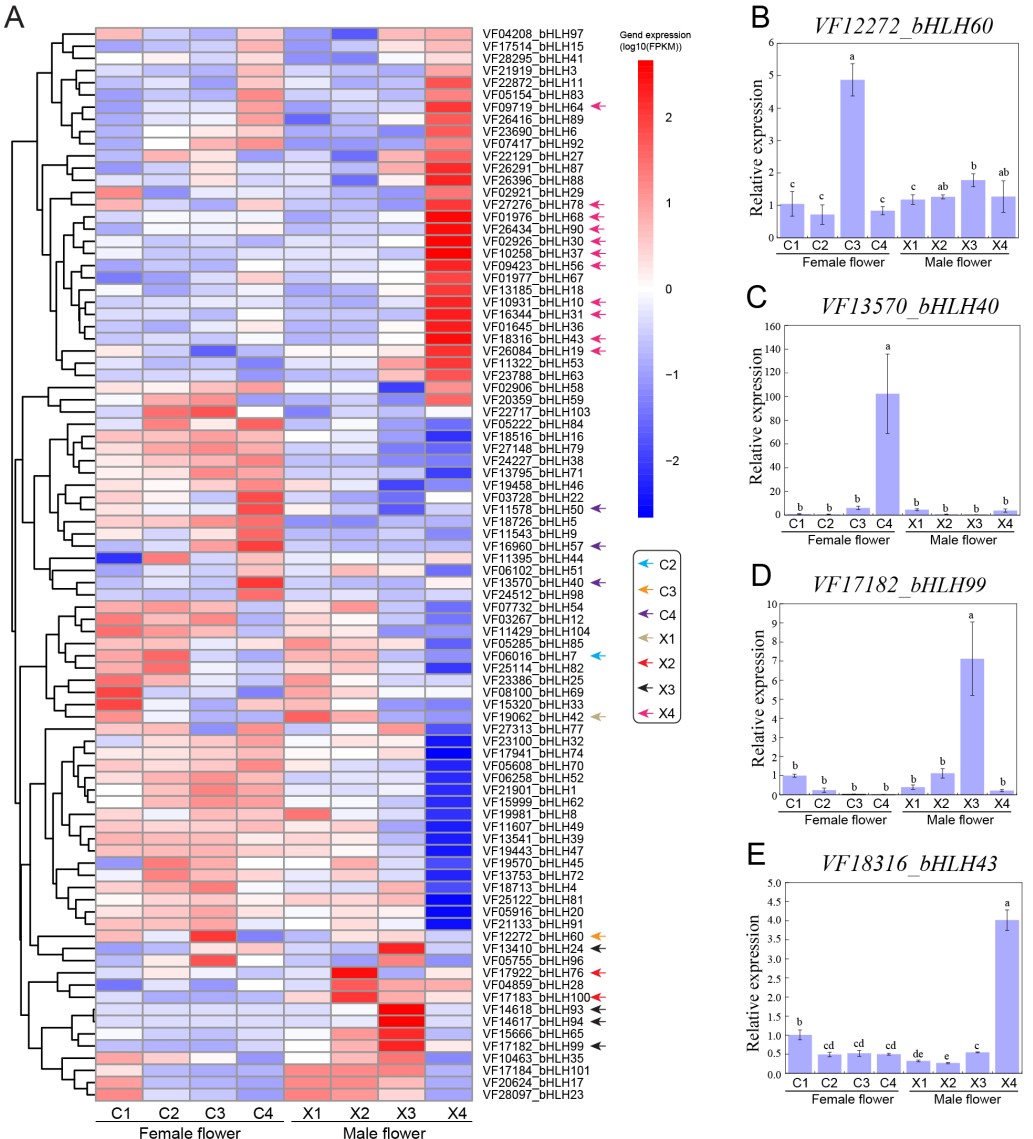

**Figure 3 Expression profiles of *VfbHLH* genes.** (A) Expression profiles of *VfbHLH* genes in male and female flowers; (B–E) Relative expression of *bHLH60*, *bHLH40*, *bHLH99*, and *bHLH43* in male and female flowers. Error bars represent the standard error of the means of three biological replicates. Different letters above the bars stand for significant differences (Tukey's multiple range tests, $P < 0.05$) between different stages.

In 10 WAF, 11 genes were considered to be specifically expressed. For example, *VfbHLH27* and *VfbHLH91*, belonging to the R subfamily, were highly expressed in 10 WAF and their FPKM values were less than one in other stages. Meanwhile, *VfbHLH62*, belonging to the L subfamily, might be associated with seed development. Additionally, there were three (*VfbHLH8/101/104*) and eight (*VfbHLH9/20/28/47/54/65/72/81*) stage-specific genes in 15 WAF and 20 WAF respectively, while there was only one (*VfbHLH60*) stage-specific gene in 25 WAF. Until 30 WAF, five genes were considered to be specifically expressed, including

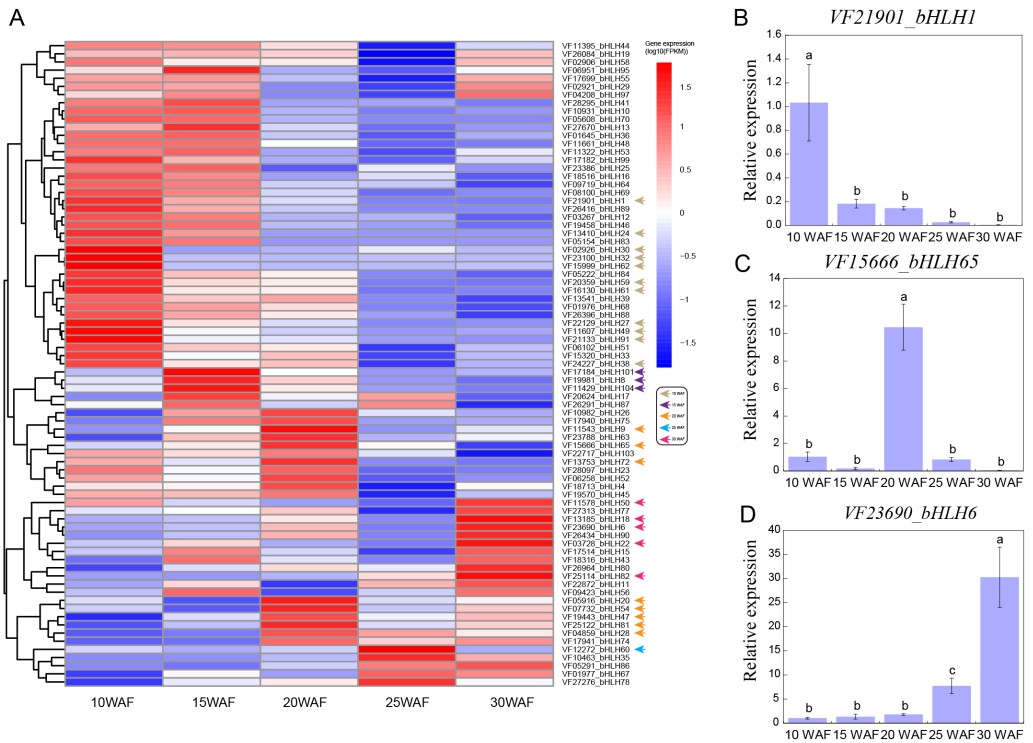

**Figure 4** **Expression profiles of *VfbHLH* genes.** (A) Expression profiles of *VfbHLH* genes; (B–D) Relative expression of *bHLH1*, *bHLH65*, and *bHLH6* in tung seed. Error bars represent the standard error of the means of three biological replicates. Different letters above the bars stand for significant differences (Tukey's multiple range tests, $P < 0.05$) between different stages.

*VfbHLH6*, *VfbHLH18*, *VfbHLH22*, *VfbHLH50*, and *VfbHLH82*, belonging to the R, N, D, Q, and A subfamily, respectively. Besides, the RT-qPCR analysis of three stage-specific genes revealed consistent expression patterns with those generated by RNA-seq data (Figs. 4B–4D). Thus, these stage-specific genes may have important roles in tung seed development and oil biosynthesis.

## Expression levels of *VfbHLH* genes in low temperature tolerance

A variety of abiotic stresses could affect a plant's health and growth, and ultimately affect the regulation of a series of stress-related genes (*Grallath et al., 2005*). Therefore, it is of great significance to clarify the regulatory pathway of stress response and grasp its regulatory factors in tung tree. At 4 °C condition, young tung tree plantlets grew normally for 2 h and 4 h, were in an extreme wilting state at 8 h and 12 h, began to wilt gradually after 72 h, and returned to normal growth at 144 h, indicating that the seedlings experienced a complex physiological change in the process of resistance to low temperature (Figs. 5A–5H). Based on cis-acting element analysis of *VfbHLH* genes, we found that 23 genes had LTR elements. We hypothesized that LTR might considerably affect the expression of 23 *VfbHLH* genes.

To better understand the stress responses involving the *VfbHLH* genes, we compared the RT-qPCR results of ten genes in different tissues and stages (Figs. 5I–5K, Table S6). In

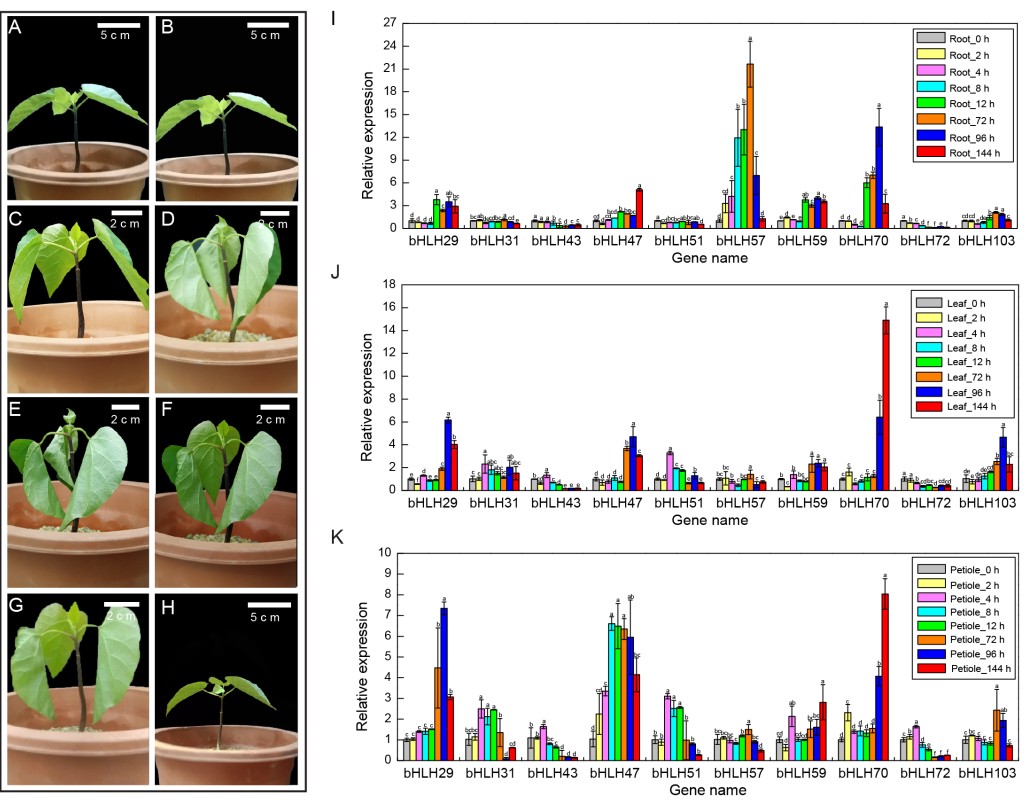

**Figure 5** **Expression levels of *VfbHLH* genes under lower temperature stress.** (A–H) Young plantlets of tung tree at 0, 2, 4, 8, 12, 72, 96, 144 h after 4 °C treatment. (I–K) Expression levels of *VfbHLH* genes in root, leaf and petiole of young plantlets after 4 °C treatment. Error bars represent the standard error of the means of three biological replicates. Different letters above the bars stand for significant differences (Tukey's multiple range tests, $P < 0.05$) between different treatment times.

roots, *VfbHLH* 29 and *VfbHLH59* were highly expressed from 0 to 144 h after the 4 °C treatment, whereas *VfbHLH57* and *VfbHLH70* expression levels peaked in less than 72 h and 96 h after the 4 °C treatment (Fig. 5I, Table S6), respectively. In leaves, *VfbHLH29*, *VfbHLH47* and *VfbHLH70* were up-regulated after 4 °C treatment (Fig. 5J, Table S6). For example, the *VfbHLH70* expression level at 144 h was approximately 15 times that before treatment. In petiole, the *VfbHLH31* and *VfbHLH51* expression levels peaked in less than 4 h, whereas *VfbHLH29*, *VfbHLH47* and *VfbHLH70* were up-regulated after the 4 °C treatment (Fig. 5K, Table S6). These results suggested that low-regulated *VfbHLH* gene expression in different tissues and stages might involve in seedling development, but the variability in their expression patterns implied that they might be functionally diverse, especially *VfbHLH29*, *VfbHLH31*, *VfbHLH47*, *VfbHLH51*, *VfbHLH57*, *VfbHLH59*, *VfbHLH70*, and *VfbHLH72*.

## DISCUSSION

The bHLH protein is the most extensive class of transcription factors in eukaryotes, which can regulate gene expression through interaction with specific motif in target genes. The

*bHLH* TF is not only universally involved in plant growth and metabolism, but also plays an important role in plant response to stress (*He et al., 2021*; *Ortolan et al., 2021*; *Qiu et al., 2020*; *Sun, Wang & Sui, 2018*). However, little is known about bHLH protein in tung tree. In the current study, we performed a genome-wide investigation of the *bHLH* gene family in tung tree, and a total of 104 *VfbHLH* genes were identified. This study provided comprehensive information on the *VfbHLH* gene family and deeper understanding of the functional divergence of *VfbHLH* genes in tung tree.

## Features of the *VfbHLH* genes in tung tree

Based on phylogenetic analysis, the *bHLH* genes of tung tree and *Arabidopsis* were divided into 18 subfamilies. In many species, genes in some clusters may expand over time. For example, rice (*Oryza sativa*) has 22 subfamilies with 167 members of *bHLH* genes, and wheat (*Triticum aestivum*) has 23 subfamilies with 225 members of *bHLH* genes (*Guo & Wang, 2017*; *Li et al., 2006*). In angiosperms, genes in subfamilies II and III(a+c)1 proposed by Pires & *Dolan (2010)* were identified as key regulators of tapetum development and male fertility (*Dolan, 2010*; *Zheng et al., 2020*). The M and I subfamilies of the tung tree were similar to subfamilies II and III(a+c)1. For example, AMS (*AtbHLH21*, *VfbHLH17*) plays a crucial role in tapetal development and the post-meiotic transcriptional regulation of microspore development (*Ferguson et al., 2017*), and DYT1 (*AtbHLH22*, *VfbHLH104*) is essential for the early development of the tapetum (*Cui et al., 2016*). Additionally, collinearity analysis showed that 14 duplicated gene pairs were collinear, and only two tandem duplications were found, which might explain why over 51.9% of *VfbHLH* genes were located on Chr1, Chr6, Chr9, and Chr10 in tung tree. Previous studies have shown that 103 genes were generated by gene duplication in tomato (*Sun, Fan & Ling, 2015*), indicating that the expansion of the tomato *bHLH* gene family was mainly driven by duplication. Therefore, gene duplications of the *bHLH* gene family in tung tree may be mainly caused by chromosome fragment replication or mass replication events.

## *VfbHLH* genes may play an important role in flower development

The functions of many bHLH proteins in plants have been studied in detail. Previous studies have suggested that *bHLH* TFs can regulate many aspects of flower development, such as, *AtSPATULA* in controlling carpel development and male sterility (*Irepan Reyes-Olalde et al., 2017*), and *SlbHLH22* in controlling flowering time (*Waseem et al., 2019*). In the female flower of tung tree, five genes (*VfbHLH7/60/40/50/57*) belonging to the L, E, Q, and J subfamilies were identified to be stage-specific genes. For example, *VfbHLH40* was highly expressed at stage 7 and had no expression in the other three stages in the female flower. *VfbHLH40* is a homologue of *MYC2* in *Arabidopsis*. In *Arabidopsis*, the *bHLH* TF *MYC2* has recently emerged as a master regulator of most aspects of the jasmonate signaling pathway (*Kazan & Manners, 2013*), and the female flower development of tung tree may be affected by jasmonic acid (JA) (*Mao et al., 2017*). These results indicated that *VfbHLH40* might play an important role in the female flower development. In addition, 18 genes were identified to be stage-specific genes in male flowers, mainly including six members of the A subfamily and three members of the J subfamily. For example, the *VfbHLH43* was

significantly higher expression at stage 7 in male flowers, which might play an important role in regulating pollen maturation.

### *VfbHLH* genes may play an important role in seed development

Tung tree is an oil crop with 50%–60% tung oil in seed (*Tan et al., 2011*). Thus, studying seed development is one of the most important aspects of tung tree. The *bHLH* family plays some roles in regulating fruit and seed development. For example, the *RETARDED GROWTH OF EMBRYO1* (*RGE1*) gene, a member of the *bHLH* family TF, and its loss-of-function mutation caused small and shriveled seeds (*Kondou et al., 2008*). *SlPRE2* is a regulator of fruit development and affects the plant response to gibberellic acid via the gibberellin pathway, and smaller seeds are observed in *SlPRE2* silenced lines (*Zhu et al., 2019*). In maize (*Zea mays*), maize opaque11 (*o11*), encoding an endosperm-specific *bHLH* TF, is a classic seed mutant with a small and opaque endosperm showing decreased starch and protein accumulation (*Feng et al., 2018*). Based on gene expression data, we identified 79 *VfbHLH* genes that had expression in five stages of tung seed. Among them, 28 genes showed stage-specific expression in different stages of tung seed. For example, *VfbHLH32*, a homologue of PACLOBUTRAZOL-RESISTANCE 1 (*PRE1*) in *Arabidopsis thaliana*, was specifically expressed at 10 WAF. Previous studies suggested PRE played a role in the PRE-IBH1-HBI1/ACEs or PRE-PAR1/PAR2/HFR1-PIF tripartite *HLH/bHLH* modules for transcriptional reprogramming, resulted in cell elongation (*Ikeda et al., 2012*). During the period of 10 WAF, the tung seed rapidly expanded. Therefore, *VfbHLH32* may regulate the cell elongation to accelerate the expansion of tung seed.

### *VfbHLH* genes are involved in the regulation of low temperature responses

Low temperature stress can greatly affect the metabolic process and transcriptional regulation mode of plants, which is mainly manifested as inhibiting the activity of various enzymes involved in multiple metabolic pathways and reprogramming the expression of related genes (*Zhu, 2016*). Low temperature is an important abiotic stress factor that affects the development of tung tree in China (*Zhang et al., 2020*). In the process of plant response to low temperature stress, *bHLH* TFs, as regulatory genes, play an important role in stress. For example, I*NDUCER OF CBF EXPRESSION1/2* (*ICE1/2*) and their homologous genes in other species, have been shown to play key roles in the response to cold stress (*Chinnusamy et al., 2003*; *Feng et al., 2012*; *Huang et al., 2013*). In addition, *PHYTOCHROME-INTERACTING FACTOR 3* (*PIF3*), a *bHLH* family TF, plays an important role in *Arabidopsis* freezing tolerance by negatively regulating the expression of genes in the C-REPEAT BINDING FACTOR (*CBF*) pathway (*Jiang et al., 2017*). In tung tree, 23 genes in the *VfbHLH* family had 31 LTR elements, which indicated that they might participate in low temperature responses. For example, *VfbHLH70*, a homologue of *ICE1* in *Arabidopsis thaliana*, was significantly up-regulated after 4 °C low temperature treatment in roots, leaves, and petioles. Through analysis of the relative expression of ten genes, we found that *VfbHLH29*, *VfbHLH31*, *VfbHLH47*, *VfbHLH51*, *VfbHLH57*, *VfbHLH59*, *VfbHLH70*, and *VfbHLH72* were important candidates in the regulation of low temperature responses.

## CONCLUSIONS

This study focused on the 104 members of the *bHLH* gene family in tung tree. Their gene structure, chromosomal distribution, phylogenetic relationship, and tissue-specific expression patterns were presented. Many *VfbHLH* genes were involved in flower and seed development and responded to low temperature stress. These results have important implications for the future functional analysis of *VfbHLHs*.

### Funding

This project was supported by the Scientific Research Fund of Hunan Provincial Education Department (grant nos. 19B600), the Changsha Municipal Natural Science Foundation (grant nos. kq2014156), and the Scientific Research and Innovation Project of Hunan Provincial postgraduates (CX20200742). There was no additional external funding received for this study. The funders had no role in study design, data collection and analysis, decision to publish, or preparation of the manuscript.

### Grant Disclosures

The following grant information was disclosed by the authors:
Scientific Research Fund of Hunan Provincial Education Department: 19B600.
Changsha Municipal Natural Science Foundation: kq2014156.
Scientific Research and Innovation Project of Hunan Provincial postgraduates: CX20200742.

### Competing Interests

Yunpeng Cao is an Academic Editor for PeerJ.

### Author Contributions

- Wenjuan Liu conceived and designed the experiments, performed the experiments, analyzed the data, prepared figures and/or tables, authored or reviewed drafts of the article, and approved the final draft.
- Yaqi Yi performed the experiments, analyzed the data, prepared figures and/or tables, and approved the final draft.
- Jingyi Zhuang performed the experiments, prepared figures and/or tables, and approved the final draft.
- Chang Ge analyzed the data, prepared figures and/or tables, and approved the final draft.
- Yunpeng Cao analyzed the data, prepared figures and/or tables, and approved the final draft.
- Lin Zhang conceived and designed the experiments, authored or reviewed drafts of the article, and approved the final draft.
- Meilan Liu conceived and designed the experiments, prepared figures and/or tables, authored or reviewed drafts of the article, and approved the final draft.

## PeerJ

## Data Availability

The data of the genome sequencing of the tung tree are available at NCBI: PRJNA503685. The RNA-seq data is available at NCBI: SRX3843588; SRS3089151; SRS3089154; SRX3843589; SRS3089148; SRS3089147; SRS3089150; SRX3843585; SRX4488507; SRX4488514; SRX4488515; SRX4488516 and SRX4488517.

## Supplemental Information

Supplemental information for this article can be found online at http://dx.doi.org/10.7717/peerj.13981#supplemental-information.

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
