# Peer review of "Genome-wide identification and transcriptional profiling of the basic helix-loop-helix gene family in tung tree (Vernicia fordii)"

_PeerJ, doi:10.7717/peerj.13981_

## Round 0.1 · original submission · Minor Revisions

Based on reviewers' comments, the authors are asked to revise the article and submit it for further process.

Reviewer 1 ·

Basic reporting

The manuscript by Liu et al. aims to perform a genome-wide identification and transcriptional analyzing of the bHLH gene family in the tung tree. The results show that (1) 104 members of VfbHLHs were identified, (2) some of the genes display stage-specific expression patterns, and (3) eight VfbHLHs appear to respond to 4℃ treatment. These results provide some basic information on the VfbHLH gene family and some candidates for future study in the tung tree. The manuscript was overall well prepared with clear figures and text. It could be considered for accept with some improvements.
(1) Abstract: line 19, adding biotic.
(2) Abstract: line 28-30, rephrase please.
(3) Abstract: line 33, change abiotic stress because low temperature could not represent all abiotic stresses, which is similar in many other sections.
(4) Introduction: Line 57-58: please give a correct reference or change the statement. Is it possible the first bHLH protein in Arabidopsis reported in 2017?
(5) Materials and methods: the parameters of most programs are missed. Please add them.
(6) Results: please carefully double check the results. Many mistakes and incorrect statements. I take some as examples, line 168-169 and figure 1, ‘I’ is likely incorrectly as ‘f’? why is the 51th residue not conserved?please check. Figure 1B: please label subfamilies name. Line 184: change ‘many’ to exact number.
(7) Figure 6 is sufficient because only few genes were tested. Please delete it.

Experimental design

no comment

Validity of the findings

no comment

Additional comments

no comment

·

Basic reporting

The manuscript was well-written with clear objectives. It cited sufficient prior publications. The figures and tables were prepared professionally. The results were substantial and convincing.

Experimental design

The manuscript clearly defined the relevant research question with identifiable knowledge gap being investigated and how to fill the gap. The experiments were conducted rigorously with advanced technology. Methods were described with sufficient information for reproducibility.

Validity of the findings

The data were made available in a public repository. The conclusions were appropriately stated with the sound results connected to the original question investigated.

Additional comments

Lines 98-104: I would like to suggest revising the objectives as the last paragraph in the “Introduction” as follows:
“The purposes of our study were to identify the tung tree bHLH gene family members, to compare their phylogenetic relationships with Arabidopsis thaliana, to analyze their gene structures, cis-regulatory elements, tissue expression patterns, as well as expression profiles under low temperature stress in young plantlets, and finally to provide new insights into understanding of molecular evolution and function of bHLH genes in tung tree. The results provide valuable clues to further reveal the role of this family in the growth and development of tung tree.”
Lines 48-49: revise the sentence as “According to the ability to bind DNA, …”
Line 57: start a separate sentence by adding a period after “)”: ). In
Lines 135, 256: change “anylysed” to “analyzed”
Line 142: delete “s” after genes in “The VfbHLH genes sequences” to “The VfbHLH gene sequences”
Lines 254, 257: delete “s” after flowers in “VfbHLHs Involved in Male and Female Flowers Development”

---

## Round 0.2 · accepted · Accept

As per the reviewers' suggestions, the paper is now acceptable.

Reviewer 1 ·

Basic reporting

The manuscript has addressed most my concerns and I don't have other suggestions.

Experimental design

Same as above

Validity of the findings

Same as above

Additional comments

Same as above

·

Basic reporting

Satisfied with my previous concerns.

Experimental design

Satisfied with my previous concerns.

Validity of the findings

Satisfied with my previous concerns.

Additional comments

Satisfied with my previous concerns.